# A Ten-Week Motor Skills Training Program Increases Motor Competence in Children with Developmental Coordination Disorder

**DOI:** 10.3390/children8121147

**Published:** 2021-12-06

**Authors:** Orifjon Saidmamatov, Quvondiq Raximov, Paula Rodrigues, Olga Vasconcelos

**Affiliations:** 1Motor Control and Learning Laboratory, CIFI2D, Centre of Research, Education, Innovation and Intervention in Sport, Faculty of Sport, University of Porto, 4200-450 Porto, Portugal; packn2@gmail.com (P.R.); olgav@fade.up.pt (O.V.); 2Faculty of Physical Culture, Urgench State University, Urgench 220100, Uzbekistan; kolya_0200.uz@mail.ru; 3Kinesiolab, Laboratory of Human Movement Analysis, Piaget Institute, 1950-157 Almada, Portugal

**Keywords:** developmental coordination disorder (DCD), motor skills training program, MABC-2

## Abstract

The present study aimed to investigate the effect of a motor skills training program in children with DCD considering their gender. The Movement Assessment Battery for Children (MABC-2) classified the children and assessed their skill changes over time. The study was implemented at four kindergartens in the Khorezm region of Uzbekistan. In the study, all the children suffered from DCD (5.17 ± 0.70 years; 10 girls), and all the indicators of MABC-2 were less than 16%. The participants were divided into an experimental group (*n* = 17), receiving ten weeks of motor skills training program for 45 min twice per week; and a control group (*n* = 7), which proceeded with exercises of everyday living. The ANCOVA showed differences between the groups in the post-test concerning each domain: manual dexterity (F _(1,_ _20)_ = 18.703, *p* < 0.001; η^2^ = 0.471); aiming and catching (F _(1,_ _20)_ = 9.734, *p* = 0.005; η^2^ = 0.317); balance (F _(1,_ _20)_ = 35.140, *p* = 0.000; η^2^ = 0.626); and total MABC-2 test score (F _(1,_ _20)_ = 66.093, *p* < 0.001; η^2^ = 0.759), with all the children in the EG exhibiting better results. The Wilcoxon test revealed statistically significant differences for the EG between moments for all the variables (*p* < 0.001) but not for the CG (*p* > 0.050). The effectiveness of the intervention program was similar across both genders. The study suggests that a 10 week motor skills training program can increase the quality of children’s motor competence and represent a valuable procedure for physical education specialists to enhance motor competence for children with DCD.

## 1. Introduction

Developmental coordination disorder (DCD) is characterized by a delay in the development of motor competence, especially the coordination of movements, which visibly impairs the child’s movements and accomplishment of daily tasks [1]. Furthermore, the feelings of inadequacy accompanying poor motor coordination are invariably reinforced through relationships with peers [2]. DCD may decrease motivation to perform physical activities and decrease opportunities to develop motor competence and fitness [3]. Rivilis et al. [3] previously observed that children with DCD, in addition to their motor problems, present lower levels of health-related physical activity. If not treated in time, the symptoms of movement problems persist in a large percentage of individuals into adulthood. Children with movement difficulties tend to be less physically active and engage in less physical activity [4].

However, spare time or recreational physical activity is essential for children and adolescents’ healthy social and physical development [5]. Consequently, DCD has received significant attention from scientists across disciplines, including pediatrics, occupational therapy, physiotherapy, kinesiology, and psychology [6]. Researchers and health professionals have developed several types of interventions focused on motor skills training programs to overcome DCD. These programs are based on daily motor skills, exercises, and activities involving fundamental motor competence. The majority of investigators studying this domain agree that these programs enhance the quality of movement and diminish DCD difficulties in children [5,7,8]. Moreover, besides improving motor competence, short-term motor skill interventions have also been shown to improve other cognitive, emotional, and psychological aspects in children with DCD [9]. Regarding the design of motor skill interventions, there have been different movement-based approaches to intervention for DCD, which are typically grouped into two wide ranges: those that utilize movement to target main performance problems, frequently defined as process-oriented approaches, and those that use activity to address the performance itself, frequently defined as task-oriented approaches [5]. In a review about the efficacy of motor interventions for children with DCD [10], the benefits for the motor performance of children with DCD over and above a lack of intervention were observed. However, task-oriented approaches were more effective than process-oriented approaches, yielding better motor competence for children with DCD. Concerning the gender effect on the efficacy of motor interventions for children with DCD, research is scarce, with only a few studies indicating equal outcomes of the effectiveness of programs on both boys and girls [11,12]. For example, Niemeijer et al. [11], using Neuromotor task training in a group of DCD seven-year-old children (20 males, 6 females), found an improvement in motor competence regardless of gender. However, gender differences and their importance for motor competence expression have been emphasized by several authors, such as Dos Santos et al. [13] and Reyes et al. [14]. The authors found that children without DCD might possess fine motor skills, while Dos Santos et al. [13] proved that inconsistent results of gross motor coordination occur with girls and Reyes et al. [14] discovered the same tendency with boys.

Scientific evidence proves that interventions are effective in the short term to develop motor competence, as well as performance in emotional, cognitive, and other psychological aspects in DCD children [9,10,11,12,13,14,15].

However, other authors, such as Nolan [16], found that girls might be superior to boys at developing postural control, with girls progressing to an ‘adult-like’ integrated open- and closed-loop strategy of controlling balance earlier than boys. These results were supported by Riach and Starkes [17], Kirshenbaum et al. [18], and Kolic et al. [19]. Nolan [16] also observed that greater balance ability and postural control were associated with Body Mass Index (BMI), with older female children and children with lower BMI being more likely to exhibit greater balance ability and postural control. Therefore, Nolan [16] highlighted the need to study the sexes separately when investigating balance in children. Reys et al. [14] supported these differences, in a study where they found that boys outperformed girls from 6 years of age to 9 years of age. In addition, children with increasing body mass index were less coordinated, while those who were stronger and more agile exhibited steeper trajectories of gross motor coordination with age.

Concerning children with DCD, Rodrigues et al. [20] systematically analyzed the differences in motor performance between genders in studies that used only the MABC-2 [21]. Their results revealed that gender differences in performance were consistent across studies, since boys demonstrated more success and ease in activities involving gross motor skills, and girls performed better in the activities involving fine motor skills. Differences in balance were not conclusive. Since few studies aimed to explore possible gender differences and the role of the environmental context in fundamental motor skill programs with children with motor problems [22], there has been less research into whether such interventions are similarly efficient between genders of children with DCD. In this scenario, if boys and girls respond differently to motor intervention programs to overcome DCD, then motor programs must be adapted, considering the differences between genders, to maximize the reduction in movement difficulties for each. Taking into consideration that there are gender differences in motor competence and task-oriented approaches are more effective, the main goal of the current study is to study the effectiveness of a motor skills training program in children with DCD and analyze possible gender differences. Thus, the hypothesis was that participation in the intervention program would significantly improve the motor competence levels of all children, with boys and girls responding differently to motor intervention.

## 2. Methods

### 2.1. Participants

With the consent of the Department of Preschool Education of the Khorezm region (Uzbekistan), from the earliest periods of the COVID- 2019 rule mitigation period, scientific research was launched among four kindergartens in the northwest region of Uzbekistan, the Urgench region. First, 63 children aged 4–6 years old with no indication of any neurological or physical impairment from these four kindergartens were tested with the MABC-2 [21]. Among the children evaluated with the MABC-2, those falling between the 5th and the 16th percentile (27 children) were selected and invited to participate in the intervention study (experimental group). However, during the pandemic time. access to one kindergarten was not accessible for intervention. Thus, we selected this kindergarten as a control group with 7 children. A total of 17 children were from the remaining kindergartens. It should be noted that 3 out of the 27 children (one of them from the experimental group and two of them from the control group) involved in the study could not participate in the post-test due to their parents’ desire that they not attend kindergarten (Figure 1). Written parental consent was obtained.

The inclusion criteria consisted of the children’s performance falling between the 5th and the 16th percentile (probable DCD (Developmental Coordination Disorder)) or below the 5th (at risk of DCD) on the total MABC-2 (Movement Assessment Battery for Children, second edition) score and having not indicated any neurological or physical impairment. Descriptive statistics on gender, age, manual preference, and MABC-2 scores can be found in Table 1.

### 2.2. Instrument

The MABC-2 test was applied and executed following the MABC-2 manual in order to measure the effects of the training program on both groups [21]. MABC-2 is an appropriate instrument with which to assess the development of the motor competencies of preschool children, and acceptable overall evidence of its validity and reliability for age band one has been found [23,24]. Smits-Engelsman et al. [24] studied the psychometric properties and the confidence of the MABC-2 in 50 typical children (3 years of age) and concluded, based on the test-retest, that the test offers a good degree of confidence, even for three-year-olds, making it sensitive at detecting individual changes. Moreover, Ellinoudis et al. [25,26], using band 1 on 183 Greek children, suggested that the MABC-2 could be a valid and reliable tool to assess children aged 3 to 5 years of age with movement difficulties. Henderson, Sugden, and Barnett [21] proposed the following cutoff points from the test manual: ≤5% atypical motor performance, which indicates DCD; 6th to 15th percentile, which means r-DCD; and any percentile higher than 16%, which indicates TD. In pre-and post-tests, each child was assessed by the same tester and recorded on video.

### 2.3. Procedures

In the task-oriented approach, the focus was on training functional tasks, which included those that involved mainly body stability (e.g., standing) and those that required body transport (e.g., hopping, skipping, running, jumping, walking, and galloping) [27]. The children in the experimental group were involved for 45 min twice a week for ten weeks. The participants were divided into four groups, corresponding to the four kindergartens, and the number of participants in each group was 3–6 children. Each training session included two main components: an initial 5 min warm-up, and 40 min of motor skill training (group training is outlined in Table 2). All the children in the experimental group completed the intervention, and all the intervention sessions were videotaped. The DCD control group and the experimental group also performed their regular classroom activities, physical education classes, and physical activities as scheduled, but only the experimental group participated in any extra training for the duration of this research. The extra training replaced the daily walk during those 45 min and was designed and taught in the indoor facilities of each school by two graduates in PE (a PHD student and a master’s degree student). They received an average of 15 h of training during the ten weeks. Due to various circumstances, such as kindergarten activities and dentist visits, several children missed one or more training sessions, which resulted in slightly different total training hours between the children in the experimental group (13.30 ± 1.02 h). Common motor problems experienced by children with DCD, such as poor agility, balance, core stability, and movement coordination, were addressed through a range of functional activities and exercises. The motor tasks were modified as the training continued to guarantee successful task execution while also offering a sufficient challenge to the child’s motor ability.

### 2.4. Data Analysis

One-way analysis of covariance (ANCOVA) was applied to examine differences in the post-test scores of all the MABC-2 domain variables between the intervention and control groups, controlling for pre-test scores [28]. The pre-test score of each domain variable was a covariant, the different groups were independent variables, and the post-test score was the dependent variable.

Next, the program effect was tested using a Wilcoxon test for repeated measure analysis in each group (EG and CG) and the Mann–Whitney was used to test gender differences within groups and moments.

The dependent variables included abilities in the domains of manual dexterity, aiming and catching, balance, and total MABC-2 scores.

The statistical analyses were performed using SPSS 26.0, and the statistical significance was assessed using an alpha level of 0.05. An estimate of the effect size for the intervention group, partial eta squared (ηp^2^), was calculated for each dependent variable. According to Cohen’s [29] guidelines, 0.0099 constitutes a small effect, 0.0588 a medium effect, and 0.1379 a large effect.

## 3. Results

The CONSORT flow diagram, illustrated in Figure 1, presents 63 children meeting the inclusion criteria. Of these, twenty-four children diagnosed with DCD by the MABC-2 program were included in the study. There were no significant differences in perinatal and social factors or motor performance between the recruited children and the non-participants. The 24 recruited children with DCD were randomized to either the experimental group (*n* = 17; mean corrected age 5.47 ± 0.514; 10 males) or the control group (*n* = 7; mean corrected age 4.71 ± 0.756; 4 males).

From pre to post-intervention, all the children from the experimental group with probable DCD or at risk demonstrated no motor impairment after intervention. Nevertheless, in the control group, the majority remained with motor impairment and even regressed, with one falling above the 16 percentiles. In the second part of the study, the MABC-2 percentile of all the children in the experimental group exceeded the 16th percentile, but the children’s performance in the control group remained virtually unchanged (Table 3).

### 3.1. Program Effect

#### 3.1.1. Manual Dexterity

The ANCOVA highlighted differences between groups in the post-test (F _(1, 20)_ = 18.703, *p* < 0.001; η^2^ = 0.471) considering the pre-test values (F _(1, 20)_ = 2.229, *p* = 0.150; η^2^ = 0.096) (see Table 4).

The Wilcoxon test revealed for the EG statistically significant differences between moments (z = 3.422, *p* < 0.001) but not for the CG (z = −1.890, *p* = 0.059) (see Table 5).

The MABC-2 indicators improved in the post-test experiment group, but the indicators decreased from the first to the second moment in the control group.

#### 3.1.2. Aiming and Catching

The ANCOVA highlighted differences between groups in the post-test (F _(1, 20)_ = 9.734, *p* = 0.005; η^2^ = 0.317) considering the pre-test values (F _(1, 20)_ = 0.245, *p* = 0.626; η^2^ = 0.012), (see Table 4). The Wilcoxon test revealed statistically significant differences between moments for the EG (z = 3.160, *p* = 0.002) but not for the CG (z = 0.420, *p* = 0.674) (see Table 5).

These effects indicated a positive overall intervention effect over time and different levels of ball skills between the groups of children.

#### 3.1.3. Balance

The ANCOVA highlighted differences between groups in the post-test (F _(1, 20)_ = 35.140, *p* = 0.000; η^2^ = 0.626) considering the pre-test values (F _(1, 20)_ = 0.039, *p* = 0.845; η^2^ = 0.002), (see Table 4).

The Wilcoxon test revealed statistically significant differences between moments for the EG (z = 3.209, *p* = 0.001) but not for the CG (z = −0.843, *p* = 0.399) (see Table 5). The performance improved more in the experimental group than in the control group from the first to the second moment.

#### 3.1.4. Total Test Score MABC-2

The ANCOVA highlighted differences between groups in the post-test (F _(1, 20)_ = 66.093, *p* < 0.001; η^2^ = 0.759) considering the pre-test values (F _(1, 20)_ = 1.129, *p* = 0.300; η^2^ = 0.051) (see Table 4).

The Wilcoxon test revealed statistically significant differences between moments for the EG (z = 3.647, *p* < 0.001) but not for the CG (z = −0.740, *p* = 0.459) (see Table 5). These results indicated a positive overall intervention effect over time, with the EG demonstrating an improved performance compared to the CG and the CG a decline in performance from the first to the second moment.

### 3.2. Gender Effect

The results of the Mann–Whitney test revealed no statistical differences between genders in both groups and moments (Table 6 and Table 7).

It was found during our study that the exposure rate equality of the intervention program administered to the children with DCD was equal for both girls and boys in both groups and in all variables (*p* > 0.05).

## 4. Discussion

The purpose of the present study was to analyze the effect of a motor skills training program on children with DCD considering their gender. As a result, the acquisition of motor skills was carefully monitored for ten weeks, and the effectiveness of the motor skills training program was assessed in a sample of children with different levels of DCD. This subject was also investigated by Farhat [30], Cacola [31], Hung [32], and Zanella [33], who studies support the use of a motor training program to increase motor competence in children with DCD. Our results indicated differences between the experimental group and the control group in all the motor competence measures, as in other studies with similar interventions with healthy [16,30], older children [34], and different genders [13,14]. Importantly, our study found that, with the exception of children in the control group, all the children with DCD improved their total test scores, manual dexterity, aiming and catching, balance, and increased percentile after a 10 week training period. Moreover, at the end of the study, it was found that not only in the MABC-2 program but also in interviews with parents and kindergarten teachers, positive changes were observed in the general physical activity of the children involved in the study. These findings suggest that a structured program of activities with an emphasis on the motor skills of manual dexterity, aiming and catching, and balance can benefit the motor competence of children with DCD, as has already been demonstrated in previous studies [32,35]. When the MABC-2 battery was used to analyze the skills in terms of manual dexterity, improvements were seen in the intervention group, which could have been due to specific tasks being performed with this skill during the intervention. However, there were no improvements in this subscale in the control group, which could have been attributed to the fact that fine motor skills require a longer acquisition time [5]. Regarding aiming and catching, the results indicate an increase in the score in each of the groups, but they were higher in the intervention group. These results agree with those of Yu et al. [36] and Woods et al. [37], which demonstrated that an intervention based on fundamental movement skills improves object control skills in pre-school children with DCD, discarding the idea that these skills are difficult to improve upon [38].

Another key finding of the study was that the aiming and catching skills of children with DCD can improve without intervention. This finding was also confirmed in other research studies [36,37]. However, it should be mentioned that the improvement rate in the aiming and catching skills was higher in the intervention group than in the non-intervention control group.

Our findings reveal that in the experimental group, specific motor competence orientation, as taught by a specialist, exerted a beneficial influence on all MACBC-2 aspects. The individual results of the experimental group demonstrated that the 17 preschoolers improved in the total percentile of the MABC-2 battery to such an extent that they could be classified in a different performance category on this test and their clinical range could be changed to normal, according to the MABC-2 manual [21]. Our results demonstrate that the experimental groups of children with lower scores on the total percentile obtained significant improvements to the point of falling within a normal motor competence range (percentile >16). Our results are in line with those of Farhat et al. [30], which demonstrated that children with serious problems in motor competence, after a structured program, benefit more than those who are framed in higher percentiles and classified with normal motor competence (that is, their possibility of improvement is greater than that of their counterparts). Concerning gender, the main finding of our study is that the results of the post-intervention program were almost identical in boys and girls. Our results are not in line with those of Bardid et al. [22] because they found that their intervention was gender-specific. These authors, using the TGMD-2, observed that boys and girls exhibited a similar gain in locomotor skills, but only the girls’ object control skills benefited from the 10 week practice.

In this study, statistically significant gender differences were found between boys and girls. All the scores and percentages of the girls were higher than those of the boys, especially in object control skills, after the 10 week intervention; the results do not support those of some other investigations [39,40,41,42], which indicate that boys generally achieve higher scores. These global differences with respect to gender could be due, in part, to the higher scores obtained by girls in manual dexterity [43,44,45] and in balance [43,46], perhaps due to the practice of stereotyped activities, and different sports activities, between boys and girls. Therefore, preschool education classes should promote co-educational sports activities to try to eliminate these gender differences.

### Limitations

This study features potential limitations. The period for determining the coordination ability of 63 children through the MABC-2 program coincided with the period when the COVID-19 quarantine restrictions were first released. Therefore, the sample size was small and the group assignment of this quasi-experimental design was not completely random for logistical reasons. This period may have led to some differences in the research results. Moreover, the COVID-19 quarantine restrictions did not allow a maintenance test. The long-term effectiveness of this intervention could not be examined, so it is suggested that in other studies this should be addressed.

## 5. Conclusions

The most important finding indicated that the motor skill-training program exerts the same positive effect on both genders. Other results from this study demonstrated that a structured 10 week motor training program based on different motor skills (e.g., manual dexterity, aiming, and catching) performed in one session of 40 min per week is more effective than traditional preschool. An education program for preschoolers with DCD led by a non-physical education teacher improves all of these skills and the overall percentile so that preschoolers who participate in intervention are no longer in the bottom percentile. This should be followed by five to one, in which they can be classified with normal motor competence. Therefore, these findings provide empirical evidence demonstrating the effectiveness of a structured motor skills training curricular program implemented in preschool education, focused on the result and not on the process.

## Figures and Tables

**Figure 1 children-08-01147-f001:**
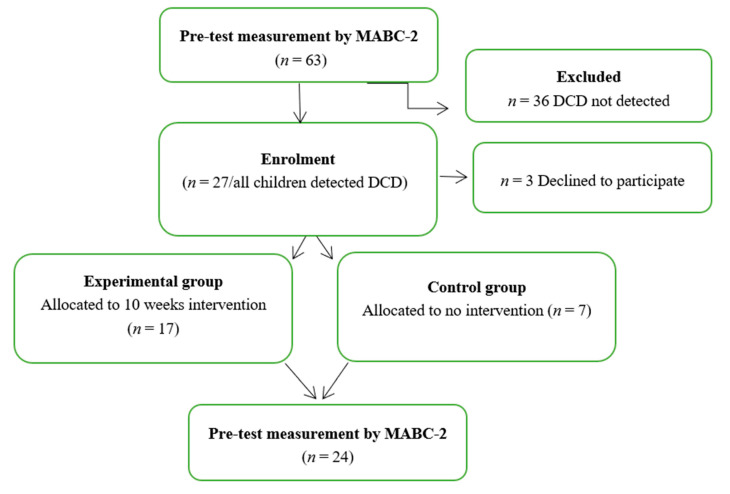
CONSORT flow diagram.

**Table 1 children-08-01147-t001:** Experimental and control groups. Characteristics of participants by age (years), sex (male; female), manual preference (right; left), and MABC-2 pre-test scores.

	Experimental Group (*n* = 17)	Control Group (*n* = 7)
Age year, mean (SD)	5.47 ± 0.514	4.71 ± 0.756
Gender (m/f)	10/7	4/3
Manual preference (R/L)	15/2	5/2
MABC-2 Total Score	5.76 ± 1.39	5.29 ± 1.11

**Table 2 children-08-01147-t002:** Motor skills training program. Examples of activities.

Domain	Examples of Activities
Manual Dexterity	Placing small beads on the side by plastic clamps Sorting and placing small products of different types with fingers Passing the lacing through the marked holes Hanging nuts on the bolts Hang the plastic clamps in the specified direction To tie clothing buttons Putting the buttons on the marked line Browsing a book and other similar games
Aiming and Catching	Catching three different balls (Robo-ball, tennis ball, football ball) Throwing three different balls to designated areas (Robo-ball, tennis ball, football ball) Bowling with twist Ping Pong Ball Catch (Get out those plastic red Solo cups and a few Ping-Pong balls) Bowling (Set up your bowling “lane” with some painter’s tape and use plastic bottles or cups for pins) Different delays and other similar games
Balance	Walk on the marked line Picking up items“Night and morning” (when “night” is said, jump into a circle, and when “morning” is said, jump out of the circle) Book balance game (Holding a book on the head without dropping it) Different delays and other similar games

**Table 3 children-08-01147-t003:** Number of children scoring in each MABC-2 percentile band at pre- and post-intervention.

Motor Difficulty Category, *n*	Experimental Group (*n* = 17)		Control Group (*n* = 7)	
	Pre	Post	Pre	Post
No motor difficulty (MABC-2 > 16th percentile)	0	17	0	1
At risk of DCD (from 6th–16th percentile)	11	0	3	1
Probable DCD (MABC-2 ≤ 5th percentile)	6	0	4	5

**Table 4 children-08-01147-t004:** Means and standard deviation for post-test MABC-2 domains by groups.

MABC-2	Group	Mean ± SD	*p*	*η* ^2^
Manual Dexterity	Experimental group Control group	7.76 ± 2.25 4 ± 1.73	0.001	0.471
Aiming and Catching	Experimental group Control group	13.9 ± 2.24 9.3 ± 3.35	0.003	0.317
Balance	Experimental group Control group	11.1 ± 1.96 6.0 ± 1.63	0.920	0.626
Total Test Score	Experimental group Control group	81.6 ± 5.911 51.43 ± 11.85	0.524	0.759

**Table 5 children-08-01147-t005:** Experimental and Control groups. Results from MABC-2 domains at pre-and post-intervention.

		Experimental Group		Control Group	
MABC-2		Mean ± SD	*p*	Mean ± SD	*p*
Manual Dexterity	Pre-test Post-test	3.82 ± 1.18 7.76 ± 2.25	0.001	4.86 ± 1.86 4.00 ± 1.73	0.059
Aiming and Catching	Pre-test Post-test	9.47 ± 2.80 13.18 ± 2.24	0.002	8.29 ± 2.69 9.29 ± 3.35	0.674
Balance	Pre-test Post-test	8.00 ± 2.34 11.12 ± 1.96	0.001	7.00 ± 2.30 6.00 ± 1.63	0.399
Total Percentile Rank	Pre-test Post-test	5.76 ± 1.39 10.65 ± 1.53	0.001	5.29 ± 1.11 4.71 ± 1.89	0.459

**Table 6 children-08-01147-t006:** Pre-intervention. Gender effect in each group at MABC-2 domains.

		Experimental Group		Control Group	
MABC-2	Gender	Mean ± SD	*p*	Mean ± SD	*p*
Manual Dexterity	Male Female	3.90 ± 1.28 3.71 ± 1.11	0.887	5.50 ± 1.73 4.00 ± 2.00	0.400
Aiming and Catching	Male Female	9.80 ± 2.48 9.00 ± 3.36	0.536	8.50 ± 0.577 8.00 ± 4.58	1.000
Balance	Male Female	8.20 ± 2.39 7.71 ± 2.43	0.669	6.75 ± 1.89 7.33 ± 3.21	1.000
Total Test Score	Male Female	58.6 ± 9.45 56.2 ± 8.57	0.536	56.7 ± 4.57 53.6 ± 8.08	0.629

**Table 7 children-08-01147-t007:** Post-intervention. Gender effect in each group at MABC-2 domains.

		Experimental Group		Control Group	
MABC-2	Gender	Mean ± SD	*p*	Mean ± SD	*p*
Manual Dexterity	Male Female	7.80 ± 2.56 7.71 ± 2.56	0.813	4.00 ± 1.82 4.00 ± 2.00	1.000
Aiming and Catching	Male Female	12.9 ± 2.07 13.5 ± 2.57	0.669	9.25 ± 3.94 9.33 ± 3.21	1.000
Balance	Male Female	11.1 ± 1.96 11.1 ± 2.11	1.000	5.50 ± 1.91 6.67 ± 1.15	0.400
Total Test Score	Male Female	81.9 ± 7.18 81.1 ± 3.93	0.887	49.5 ± 16.0 54.0 ± 4.35	0.629

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
