# Peer review of "A Ten-Week Motor Skills Training Program Increases Motor Competence in Children with Developmental Coordination Disorder"

_children, 2021, doi:10.3390/children8121147_

Round 1

Reviewer 1 Report

The aim of the present study was to investigate the effect of a motor skills training program in children with DCD considering their gender. The Movement Assessment Battery for Children (MABC-2) was used to classify the children and to assess skill changes over time. The study was implemented at four kindergartens in the Khorezm region of Uzbekistan. All 24 children (5.17±0.70 years old; 10 girls) with DCD who taken part were referred to the research and scored at and below the 16 the percentile for their age on MABC-2. Participants were divided into an experimental group (n=17) receiving 10 weeks of motor skills training program for 45 min twice per week and the control group (n=7) proceeded with exercises of everyday living. All children in the experimental group had a significant increase in total percentile rank of MABC-2 and concerning each domain (manual dexterity; throwing and catching; balance). In the control group, a significant decrease in the total percentile rank of MABC-2 and in each domain of MABC-2 was observed. The effectiveness of the intervention program was similar across both genders. The study supports that a period of 10 weeks of a motor skills training program can have an increase in the quality of children’s motor coordination and can represent a useful procedure for physical education specialists to enhance motor skills for children with DCD.

The format of the citations is incorrect according to the rules of the research journal.

Title

Motor coordination should be replaced by motor competence which is what the MABC-2 battery measures and change it throughout the entire document

Abstract

Motor Skills should be replaced by Motor Competence. Authors have to do that in the hole manuscript.

The intervention and control group sample is too small in order to interpret properly the results. I consider this work suitable as a pilot test. A larger sample should be collected to allow transferable conclusions to be reached.

 Authors must indicate the p value in each case (i.e. Manuel dexterity, aiming and catching and balance).  Effect size should be reported in the Abstract too.

Introduction

Authors are recommended to introduce more current scientific evidence to elaborate this section. Children and other MDPI Journals published studies in this sense throughout 2021. For example:

Honrubia-Montesinos, C.; Gil-Madrona, P.; Losada-Puente, L. Motor Development among Spanish Preschool Children. Children 2021, 8, 41

Navarro-Patón, R.; Brito-Ballester, J.; Villa, S.P.; Anaya, V.; Mecías-Calvo, M. Changes in Motor Competence after a Brief Physical Education Intervention Program in 4 and 5-Year-Old Preschool Children. Int. J. Environ. Res. Public Health 2021, 18, 4988. https://doi.org/10.3390/ijerph18094988

 Navarro-Patón, R.; Martín-Ayala, J.L.; Martí González, M.; Hernández, A.; Mecías-Calvo, M. Effect of a 6-Week Physical Education Intervention on Motor Competence in Pre-School Children with Developmental Coordination Disorder. J. Clin. Med. 2021, 10, 1936. https://doi.org/10.3390/jcm10091936

Navarro-Patón, R.; Pueyo Villa, S.; Martín-Ayala, J.L.; Martí González, M.; Mecías-Calvo, M. Is Quarter of Birth a Risk Factor for Developmental Coordinator Disorder in Preschool Children? Int. J. Environ. Res. Public Health 2021, 18, 5514. https://doi.org/10.3390/ijerph18115514

Research hyotheses have not been formulated

Method

Psychometric properties of the MABC-2 test are missing. It is need to add the psychometric of the MABC-2

Line 89-93: The inclusion criteria consisted of children´s performance falling between the 5th and 89 the 16th percentile (probable DCD) or below the 5th (at risk of DCD) on the total score of 90 MABC-2 and having not indicated of any neurological or physical impairment. Descrip-91 tive statistics…. How many children have been tested to get this sample?

Line 137: “M-ABC”

The tool must be cited in the same way throughout the manuscript

Discussion

in this section, you are not studying the gender effect. In order to do this, a regression analysis is needed. What you are providing are the differences between the gender of participants. Moreover, as I have said before, the sample is too small to make gender differences. Results could not be extrapolated.

The discussion is poor. The authors present a general and obvious idea, but do not work the data obtained in a detailed way, comparing them with those of other previous investigations. In fact, the discussion and the conclusion say the same thing.

References

References are arranged alphabetically which causes them to get out of order in the text and not follow the order of appearance.

Author Response

Dear Reviewer,

The authors worked on the provided comments and sharing the responses, attached, please.

Best regards,

Authors

Reviewer 2 Report

Manuscript Number: children-1450622

Title: A ten-week motor skills training program increases motor coordination in children with Developmental Coordination Disorder

General Comments:

Thank you so much for the opportunity to review this manuscript. This project fills in some gaps on understanding the effect of motor intervention on motor performance in children with DCD. However, the rationale for the study, especially the gender effect, was not clearly laid out. Several importance details on the methods were not descripted. Overall, the current manuscript needs more improvement and the current version is not publishable.

Introduction:

  1. Page 2, line 54-55: “….yielding better functional performance results in less time for children with DCD”. I am not so sure what this sentence means here. Do you suggest that “better performance results in faster speed”? Could you clarify?
  2. Page 2, between paragraphs 1 and 2: It needs some transitions from the previous paragraph (mainly on intervention) to the next (suddenly start to talk about gender differences).
  3. Page 2, line 56-60: Some parts of the sentences are broken. Could you please re-write this part?
  4. Page 2, 2nd paragraph: Could you explain more on those studies that reported gender differences? What were the age ranges for those children? Which types of trainings were used? How did these trainings relate to your previous discussions on different approaches?
  5. Page 2, end of the introduction: could you explain more about the “motor skills training program”. How does this training program relate to the discussion on task-oriented versus process-oriented approaches? Why do you select this program? What are your predictions?

Methods:

  1. The ratio between the qualified “DCD” and recruited sample was unbelievably high (27/63~=43%). It is known that the average occurrence of DCD is roughly about 6%. Could you explain why almost half of your recruited sample is qualified as DCD (including probable DCD)?
  2. Page 4, 2.3 Procedure: Did the experimental group have extra 45 minutes motor training? If all children have the same regular schedule, what did the control group do during those 45 minutes?
  3. Who were implementing that training program? How those activities were introduced/arranged in sequence?
  4. Was it a double-blind study? Who were those evaluators in the study?

Results:

  1. Page 5: what was the inclusion criteria for those 63 recruited children?
  2. Page 5: If all children were randomly selected to two groups, why could one group include 17 children while the other only included 7 children? Please report the effect size/power of those analyses.
  3. It was quite surprising that all children in the experimental group show no motor impairment. This brings me back the question on the inclusion criteria for the study.

Discussion:

  1. P7: 2nd paragraph in discussion: how was the current motor program different from those in the literature? I am not sure the connection between those motor activities in the study and the term “physical activity”. There was no measure on physical activity in this study.
  2. P8: limitation. The small sample size, especially 7 children in the control group needs to be mentioned.

Author Response

Dear reviewer

Please find our response below and I am sharing it, attached, please.

Best regards

Authors

Round 2

Reviewer 1 Report

The manuscript have been improved, however some questions must be improve too. For example,

  1. The format of the journal has not been followed. For example, see Title format.
  2. It is difficult to follow the argument of the manuscript as the changes have been made.
  3. Revise must be revised in line 73-74. line 224
  4. Authors must indicate in the introduction what Scientific evidence indicates about interventions effective in the short term to improve Motos Competence in children with DCD.
  5. The figure 1 is no clear. It must be improved
  6. Results: How can the authors explain that manual dexterity and balance decreased in the control group? (table 5)
  7. Please elaborate in your discussion in more detail. There may be a problem since your intervention is clearly focused on the outcome measures. Is there a risk, that your children have adapted to the demands of the tests but not to general competencies?  
  8. The discussion should be expanded by including the results of other similar investigations in Children with DCD (line 349). For example:

Navarro-Patón, R.; Martín-Ayala, J.L.; Martí González, M.; Hernández, A.; Mecías-Calvo, M. Effect of a 6-Week Physical Education Intervention on Motor Competence In Pre-School Children with Developmental Coordination Disorder. J.Clin.Med. 2021, 10, 1936. https://doi.org/10.3390/jcm10091936

Yu, J.J.; Burnett, A.F.; Sit, C.H. Motor Skill Interventions in ChildrenWith Developmental Coordination Disorder: A Systematic Review and Meta-Analysis. Arch. Phys. Med. Rehabil. 2018, 99, 2076–2099.

Caçola, P.; Romero, M.; Ibana, M.; Chuang, J. Effects of two distinct group motor skill interventions in psychological and motor skills of children with Developmental Coordination Disorder: A pilot study. Disabil. Health J. 2016, 9, 172–178

Another concern is a lack of a maintenance test.  The effectiveness of this intervention is much less compelling without examining long term benefits.

I hope these comments help to improve the manuscript.

Best regards

Author Response

A dear commenter, Thank you for your feedback on improving the quality of the article.
I made changes based on your comments. 
If you have again comments, please let me know. I will correct them and resend them.

best regards

Orfijon

This manuscript is a resubmission of an earlier submission. The following is a list of the peer review reports and author responses from that submission.

Round 1

Reviewer 1 Report

1.- The format of the citations is incorrect according to the rules of the research journal.

2.- The control group sample is too small in order to interpret properly the results.

3.- Psychometric properties of the MABC-2 test are missing.

4.- Data analysis applied are not suitable. In fact, this section is poorly described. To my knowledge, ANOVA is more suitable used to compare more than 2 groups. Therefore, data analysisi should be changed. Moreover, you do not explainn other measures you have included in the study. Everything should be reflected in data analysis section.

5.- Paragraph 211: in this sections you are not stydyimg the gender effect. In order to do this, a regression analysis is needed. What you are providing are the differences between the gender of participants. Moreover, as i have said before, the sample is too small to make gender differences. Results could not be extrapolated.

6.- The discussion is poor. The authors present a general and obvious idea, but do not work the data obtained in a detailed way, comparing them with those of other previous investigations. In fact, the discussion and the conclusion say the same thing.

7.- I consider this work suitable as a pilot test. A larger sample should be collected to allow transferable conclusions to be reached. In any case, I think they should try to publish this work in a research journal with less impact.

Author Response

Response to Reviewer 1 Comments

Point 1: The format of the citations is incorrect according to the rules of the research journal.

Response 1: Updated properly in the paper

Point 2: The control group sample is too small in order to interpret properly the results.

Response 2: This comment is included in the limitations paragraph accordingly: ADDED: Limitationsà the study has potential limitations. The period of determining the coordination ability of 63 children through the MABC-2 program coincided with the period when the COVID-19 quarantine restrictions were first released. This period may have led to some differences in research results that can be considered the research's limitation. Moreover, the COVID-19 quarantine restrictions did not allow a maintenance test.  The long-term benefits effectiveness of this intervention could not be examined so it is suggested that in other studies this should be addressed.

Point 3: Psychometric properties of the MABC-2 test are missing

Response 3: Psychometric properties of the MABC-2 test are included in the paper: ADDED:  Instrumentà It was applied the MABC-2 test, executed following the MABC-2 manual, to measure the effects of the training program on both groups [19]. MABC-2 has been an appropriate instrument to assess the development of motor competencies of preschool children, and acceptable overall evidence of validity and reliability of the MABC-2 for age band one has been found [20, 37]. The authors propose the following cutoff points from the test manual ≤ 5% atypical motor performance, which indicates DCD; 6th to 15th percentile, which means r-DCD; and any percentile higher than 16 percent, which indicates TD.

Point 4: Data analysis applied is not suitable. In fact, this section is poorly described. To my knowledge, ANOVA is more suitable used to compare more than 2 groups. Therefore, data analysis should be changed. Moreover, you do not explain other measures you have included in the study. Everything should be reflected in data analysis section.

Response 4: Data analysis part is improved and updated as per your suggestions. UPDATED:

Data analysisà One-way analysis of covariance (ANCOVA) was applied to examine differences in post-test scores of all M-ABC domain variables among intervention and control groups controlling for pre-test scores [19]. The pre-test score of each domain variable was a covariant, the different groups were independent variables, and the post-test score was the dependent variable.

Then, the program effect was tested using a Wilcoxon test for repeated measure analysis in each group (EG and CG) and Mann-Whitney to test gender differences within groups and moments.

The dependent variables included the abilities in the domains of manual dexterity, aiming and catching, balance, and total MABC-2 scores.

Statistical analyses were performed using SPSS 26.0, and statistical significance was assessed using an alpha level of .05. An estimate of the effect size for the intervention group, partial eta squared (ηp2), was calculated for each dependent variable. According to Cohen’s [11] guidelines, 0.0099 constitutes a small effect, 0.0588 a medium effect, and 0.1379 a large effect.

The Wilcoxon test revealed for the EG statistically significant differences between moments (Z= 3.422, p = 0.001) but not for the CG (Z= -1.890, p = 0.059) (see table 5).

MABC 2 indicators improved in the post-test experiment group, but the indicators decreased from the first to the second moment in the control group.

Aiming and Catching 

The ANCOVA showed differences between groups in the post-test (F (1, 20) = 9.734, p= 0.005; η2 = 0.317) considering the pre-test values (F (1, 20) = 0.245, p= 0.626; η2 = 0.012), (see table 4). The Wilcoxon test revealed for the EG statistically significant differences between moments (Z= 3.160, p = 0.002) but not for the CG (Z= 0.420, p = 0.674) (see table 5).

These effects indicated a positive overall intervention effect over time and different levels of ball skills between groups of children.

Balance

The ANCOVA showed differences between groups in the post-test (F (1, 20) = 35.140, p= 0.000; η2 = 0.626) considering the pre-test values (F (1, 20) = 0.039, p=0.845; η2 = 0.002), (see table 4).

The Wilcoxon test revealed for the EG statistically significant differences between moments (Z= 3.209, p = 0.001) but not for the CG (Z= -0.843, p = 0.399) (see table 5). The performance improved in the experimental group than the control group from the first to the second moment.

Total Test Score MABC-2

The ANCOVA showed differences between groups in the post-test (F (1, 20) = 66.093, p= 0.000; η2 = 0.759) considering the pre-test values (F (1, 20) = 1.129, p=0.300; η2 = 0.051) (see table 4).

The Wilcoxon test revealed for the EG statistically significant differences between moments (Z= 3.647, p = 0.000) but not for the CG (Z= -0.740, p = 0.459) (see table 5). These results indicated was a positive overall intervention effect over time, with the EG showing an improved performance compared to the CG and a decline in the performance for the CG from the first to the second moment.

Point 5: Paragraph 211: in this sections, you are not studying the gender effect. In order to do this, a regression analysis is needed. What you are providing are the differences between the gender of participants. Moreover, as I have said before, the sample is too small to make gender differences. Results could not be extrapolated.

Response 5: Since there is no difference between the two sexes, the data are commonly analyzed. Added: Gender effect partà The results of the Mann-Whitney showed no statistical differences between sexes in both groups and moments (table 6 and table 7).

Point 6: The discussion is poor. The authors present a general and obvious idea, but do not work the data obtained in a detailed way, comparing them with those of other previous investigations. In fact, the discussion and the conclusion say the same thing.

Response 6: Authors worked on your valuable comments and enriched the content of the conclusion part. ADDED: 

The purpose of the present study was to analyze the effect of a motor skills training program in children with DCD considering their gender. For this reason, the motor skill acquisition was carefully monitored for ten weeks, and the effectiveness of the motor skills training program was examined in a sample of children with different levels of DCD. Importantly, our study confirmed that except for children in the control group, all children with DCD displayed improvement after a 10-week training period.

Inspection of the intervention group's findings indicated that 17 children (100%) improved to the point where they could be categorized in a different performance category on the MABC and moved from the clinical to the normal range.

The main finding of our study is that the results of the post-study MABC-2 program of girls and boys in the intervention group were almost identical. However, it was observed to have a more negative effect on boys than girls without intervention in a 10-week study period.

 Involvement of teachers, parents, and physical education is likely to ensure that the learned skills will continue to be used after the formal intervention and to maximize transfer into daily life, and ensure longer-term progress [4]. More studies are needed to study the most effective strategy that parents and teachers can be engaged in instruction.

Point 7: I consider this work suitable as a pilot test. A larger sample should be collected to allow transferable conclusions to be reached. In any case, I think they should try to publish this work in a research journal with less impact.

Response 7: I highly appreciate your valuable advice on my research paper. I would highly appreciate it if you consider this updated version where the authors did a huge input to update the paper as per your remarkable suggestions. This pilot study was conducted in Uzbekistan together with the supervision of Portugal professors which supports the internationalization of the research on sport and medical sciences. At the same time, the conducted research supports the sustainable development goals by ensuring reduced inequalities (SDG 10) and good health (SDG 3) both in developed countries (Portugal) and developing countries (i.e., Uzbekistan).I truly believe that you will consider my recent input to update the paper and I hope that you will support publishing this manuscript at this journal.

Reviewer 2 Report

The study presents a quasi-experiment evaluating a PE intervention to foster Motor Competence. N = 24 preschool children with DCD participated at the controlled study. The intervention group participated at an intervention led by a graduate in PE instead of “regular” PE classes. Results demonstrate positive effects on different outcome measures (with a focus on the skills “manual dexterity”, “aiming and catching”, and “balance” that were assessed with the MABC-2). The theme is highly interest for researched in the field of early childhood education, focusing specifically on the analysis of motor skills of a group of children, 

This article presents as weaknesses, a non-random sample and the intervention approach of the program is not well defined; but also a strength, a quasi- experimental design and an intervention program with children with DCD.

However, before publication one major and a few minor issues should be considered by the authors (Please see attached document).

Majors concern:

Please use a more appropriate approach for your statistical analyses. My main concern is that you did not consider the pre-test values for the analyses of the post-test values although you observe some significant differences in the pre-test. Even for non-significant differences between CG and IG in the pre-test you should consider that due to your low statistical test power (i.e., small sample size), there may be relevant (although non-significant) pre-test differences influencing the post-test results. Thus, from my perspective, ANCOVA or MANOVA would have been more appropriate approaches. 

The main concerns are around the description of the methodology which must be very clear for an intervention study. 

Another major concern is a lack of a maintenance test.  The effectiveness of this intervention is much less compelling without examining long term benefits.  I highly recommend assessing each group in a follow-up test.

Finally, please also consider more in depth the variety of factors that may also relate to the performance of motor skills, such as size, previous experience (nurture vs. nature), etc. and potentially sharing more the applicability of your findings - such as how would an early childhood educator use this information.

Citations throughout the document do not correspond to the journal’s citation rules. Authors are requested to modify this aspect throughout the document (please use MDPI and ACS Style)

Form minor concern, please see the attached document

Author Response

Response to Reviewer 2. Comments

Point 1: Please use a more appropriate approach for your statistical analyses. My main concern is that you did not consider the pre-test values for the analyses of the post-test values although you observe some significant differences in the pre-test. Even for non-significant differences between CG and IG in the pre-test, you should consider that due to your low statistical test power (i.e., small sample size), there may be relevant (although non-significant) pre-test differences influencing the post-test results. Thus, from my perspective, ANCOVA or MANOVA would have been more appropriate approaches. 

Response 1: Statistical analyses part is improved and updated as per your suggestions.

UPDATED:

Manual dexterity

The ANCOVA showed differences between groups in the post-test (F (1, 20) = 18.703, p= 0.000) considering the pre-test values(F (1, 20) = 2.229, p= 0.150) (see table 4).

The Wilcoxon test revealed for the EG statistically significant differences between moments (Z= 3.422, p = 0.001) but not for the CG (Z= -1.890, p = 0.059) (see table 5).

MABC 2 indicators improved in the post-test experiment group, but the indicators decreased from the first to the second moment in the control group.

Aiming and Catching 

The ANCOVA showed differences between groups in the post-test (F (1, 20) = 9.734, p= 0.005; η2 = 0.317) considering the pre-test values (F (1, 20) = 0.245, p= 0.626; η2 = 0.012), (see table 4). The Wilcoxon test revealed for the EG statistically significant differences between moments (Z= 3.160, p = 0.002) but not for the CG (Z= 0.420, p = 0.674) (see table 5).

These effects indicated a positive overall intervention effect over time and different levels of ball skills between groups of children.

Balance

The ANCOVA showed differences between groups in the post-test (F (1, 20) = 35.140, p= 0.000; η2 = 0.626) considering the pre-test values (F (1, 20) = 0.039, p=0.845; η2 = 0.002), (see table 4).

The Wilcoxon test revealed for the EG statistically significant differences between moments (Z= 3.209, p = 0.001) but not for the CG (Z= -0.843, p = 0.399) (see table 5). The performance improved in the experimental group than the control group from the first to the second moment.

Total Test Score MABC-2

The ANCOVA showed differences between groups in the post-test (F (1, 20) = 66.093, p= 0.000; η2 = 0.759) considering the pre-test values (F (1, 20) = 1.129, p=0.300; η2 = 0.051) (see table 4).

The Wilcoxon test revealed for the EG statistically significant differences between moments (Z= 3.647, p = 0.000) but not for the CG (Z= -0.740, p = 0.459) (see table 5). These results indicated was a positive overall intervention effect over time, with the EG showing an improved performance compared to the CG and a decline in the performance for the CG from the first to the second moment.

Gender effect

The results of the Mann-Whitney showed no statistical differences between sexes in both groups and moments (table 6 and table 7).

Point 2: The study presents a quasi-experiment evaluating a PE intervention to foster Motor Competence. N = 24 preschool children with DCD participated at the controlled study. The intervention group participated at an intervention led by a graduate in PE instead of “regular” PE classes. Results demonstrate positive effects on different outcome measures (with a focus on the skills “manual dexterity”, “aiming and catching”, and “balance” that were assessed with the MABC-2). The theme is high interest for research in the field of early childhood education, focusing specifically on the analysis of motor skills of a group of children, 

Response 2: The DCD control group and the experimental group also performed their regular classroom activities, physical education classes, and physical activities as scheduled, but only the experimental group participated in any extra training for the duration of this research.

Point 3: This article presents as weaknesses, a non-random sample and the intervention approach of the program is not well defined; but also a strength, a quasi-experimental design and an intervention program with children with DCD.

Response 3: The intervention approach part is improved and updated as per your suggestions.

UPDATED:

In the task-oriented approach, the focus was on training functional tasks, which included those that involved mainly body stability (e.g., standing) and those that required body transport (e.g., hopping, skipping, running, jumping, walking,   and galloping) [10]. The children in the experimental group were involved for 45 minutes twice a week for ten weeks. The participants were divided into four groups corresponding to the four kindergartens, and the number of participants in each group was 3-6 children. Each training session included two main components: initial, 5-min warm-up, and a 40-min motor skill training (group training are outlined in Table 2). All children of the experimental group completed the intervention, and all intervention sessions were videotaped. The DCD control group and the experimental group also performed their regular classroom activities, physical education classes, and physical activities as scheduled, but only the experimental group participated in any extra training for the duration of this research. They received an average of 15 hours of training during the ten weeks. Due to various circumstances like kindergarten activities and dentist visits, several children missed one or more training sessions, which resulted in slightly different total training hours between the children in the experimental group (13.30 ± 1.02 hours). Common motor problems experienced by children with DCD, such as poor agility, balance, core stability, and movement coordination, were addressed through a range of functional activities and exercises. The motor tasks were modified as the training continued to guarantee successful task execution while also offering a sufficient challenge to the child's motor ability.

Round 2

Reviewer 2 Report

Although the authors have made an effort to improve the manuscript, I have several doubts regarding the possible biases of the research that make it impossible for me to provide an opinion for this manuscript to be published. For example, the study has been carried out during confinement due to COVID-19; and this has been able to influence the results: For example in the decrease of motor competition of preschoolers of the control group. The results were also influenced by the 1-year difference between children in the experimental group and preschool children in the control group. This age difference means that by simple maturation older preschoolers achieve higher scores.

In addition, errors have been detected that have already been reported in the previous review and have not been corrected. For example, the authors were told that 1/3 of the references are less than 5 years old, currently only 27 is(13 of 41). It also appears that the authors have not reviewed the journal’s rules. References 2 and 18 appear in line 40. This is complicated as there is no further reference in the middle of the text. This may be because the authors continue to cite in the order in which they have placed the bibliographic references (The authors follow the APA rules; they order by surname). In addition, the numbering of the citations in the text does not match the numbering in the references (

(for example: line 303 Farhat et al. 13; for reference 13, this corresponds to De Milander et al., 2014 according to the authors).
Sele has provided current references that have not been used in the manuscript.

In addition, new errors have been found that may have arisen from the authors' failure to carry out a thorough review.

Therefore, the manuscript cannot be accepted.